# Gradient-Free Adversarial Training Against Image Corruption for Learning-based Steering

**Yu Shen**
University of Maryland
College Park, MD 20742
yushen@umd.edu

**Laura Zheng**
University of Maryland
College Park, MD 20742
lyzheng@umd.edu

**Manli Shu**
University of Maryland
College Park, MD 20742
manlis@umd.edu

**Weizi Li**
University of Memphis
Memphis, TN 38152
wli@memphis.edu

**Tom Goldstein**
University of Maryland
College Park, MD 20742
tomg@cs.umd.edu

**Ming C. Lin**
University of Maryland
College Park, MD 20742
lin@cs.umd.edu

## Abstract

We introduce a simple yet effective framework for improving the robustness of learning algorithms against image corruptions for autonomous driving. These corruptions can occur due to both internal (e.g., sensor noises and hardware abnormalities) and external factors (e.g., lighting, weather, visibility, and other environmental effects). Using sensitivity analysis with FID-based parameterization, we propose a novel algorithm exploiting basis perturbations to improve the overall performance of autonomous steering and other image processing tasks, such as classification and detection, for self-driving cars. Our model not only improves the performance on the original dataset, but also achieves significant performance improvement on datasets with multiple and unseen perturbations, up to 87% and 77%, respectively. A comparison between our approach and other SOTA techniques confirms the effectiveness of our technique in improving the robustness of neural network training for learning-based steering and other image processing tasks.

## 1   Introduction

Autonomous driving is a complex task that requires many software and hardware components to operate reliably under highly disparate and often unpredictable conditions. In this work, we study "learning-based steering" as it contains both perception and control, both critical components for autonomous driving. While on the road, vehicles are going to experience day and night, clear and foggy conditions, sunny and rainy days, as well as bright cityscapes and dark tunnels. All these external factors in conjunction with internal factors of the camera (e.g., those associated with hardware) can lead to quality variations in input data for image-based learning algorithms. One can harden machine learning systems to these degradations by simulating them at training time [6]. However, an algorithmic tool for analyzing the sensitivity of real-world neural network performance on the properties of (and corruptions to) training images is lacking. More importantly, a mechanism to leverage such a sensitivity analysis for improving learning outcomes needs to be developed. In this work, we quantify the influence of image quality on the task of "learning to steer," study how training on degraded and low-quality images can boost robustness to image corruptions, and provide a systematic approach to improve the performance of learning algorithms using quantitative analysis.

Image degradations can be simulated by varying attributes such as blur, noise, distortion, color representations (such as RGB or CMY) hues, saturation, and intensity values (HSV). However, identifying the correct combination of the simulated corruptions to obtain optimal performance on

35th Conference on Neural Information Processing Systems (NeurIPS 2021).

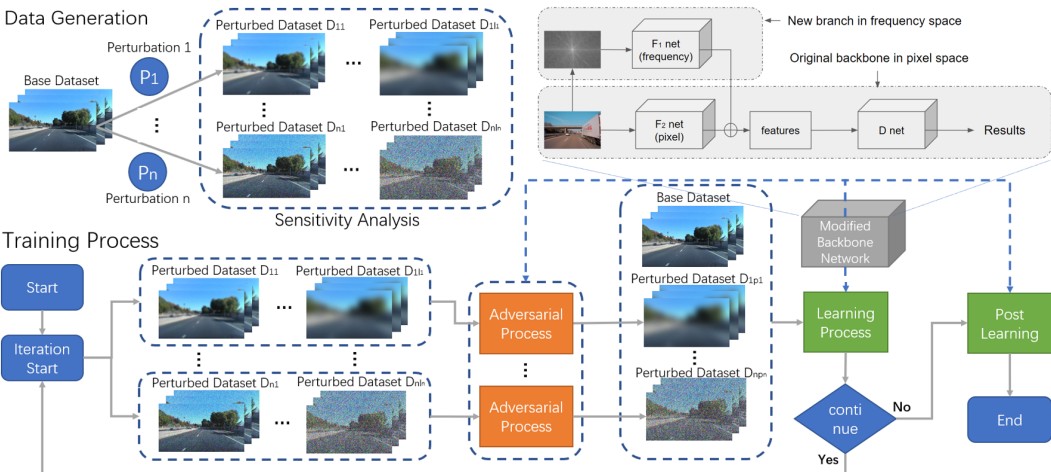

Figure 1: Pipeline of our method. **Data generation**. We generate perturbed datasets of each factor at multiple levels based on the FID-parameterized sensitivity analysis results. **Training process**. First stage: in each iteration, we first augment the training dataset with "adversarial images" generated by applying an image corruption; we then combine the base and perturbed datasets to train our model to maximize the overall performance. A frequency-space branch is added to the backbone when frequency-related perturbations (e.g., blur, noise) need to be handled. Second stage: in post-learning, the model is fine-tuned solely on clean data to boost accuracy, while performing validation on both clean and perturbed data with an early break when overall performance decreases to maintain the performance on the perturbed data.

real data is a difficult—if not impossible—task, as it requires domain transfer and exploring a high dimensional parameterized space.

We *design a systematic method for measuring the severity of image degradation and predicting the impact of such degradation on model performance*. Inspired by the use of image feature variance in sensitivity analysis [33], we measure the difference between real-world image distributions and simulated/degraded image distributions using Fréchet Inception Distance (FID). Our results confirm that FID can help predict the performance of a model trained using simulated data and deployed in the real world. Next, we use FID between different simulated datasets as a unified metric to parameterize the severity of various image degradations due to different factors.

Borrowing concepts from the adversarial attack literature [28, 35, 43], we *build a scalable training scheme for enhancing the robustness of autonomous driving against various combinations of image degradations, while* increasing *the overall accuracy of the steering task on clean data*. Our proposed method constructs a dataset of adversarially degraded images by applying optimization within the space of possible degradations during training. As shown in Fig. 1, the method begins by training on a set of real and simulated/degraded images using arbitrary degradation parameters. During each training iteration, the parameters are updated to generate a new degradation set so that the model performance is (approximately) minimized. The network is then trained on these adversarially degraded images to promote robustness. A post-training step is applied to further improve the performance on clean data without weakening robustness. Our proposed algorithm uses our FID-based parameterization to discretize the search space of degradation parameters and accelerates the process of finding optimal parameters.

Experiments show that our algorithm improves the performance of "learning to steer" up to **97%** in mean accuracy over baselines, and especially improves the performance on datasets contaminated with complex combinations of perturbations (up to **87%**). It additionally boosts the test performance on degradations that are not seen during training, including simulated snow, fog, and frost (up to **77%**). We also compare our approach with other SOTA techniques (e.g., data augmentation and adversarial training) on visual processing tasks such as detection and classification. Our method consistently achieves higher performance. In addition, our method is easy to implement and can be readily integrated with other frameworks such as object detection, classification, regression, etc.

Finally, we propose a comprehensive robustness evaluation standard under four different scenarios: clean data, single-perturbation data, multi-perturbation data, and previously unseen data. While

state-of-the-art studies usually conduct testing under one or two scenarios (e.g., ImageNet-C [20]), our work tests and verifies results under four meaningful scenarios. We plan to release code and datasets for benchmarking "autonomous driving under perturbations" using unseen factors such as image corruptions in ImageNet-C [20], totaling **480** datasets and **26M** images.

## 2 Related Work

The influence of noise and distortion on real images for learning tasks has been well explored. For example, researchers have examined the impact of optical blur on convolutional neural networks and present a fine-tuning method for recovering lost accuracy using blurred images [41]. This fine-tuning method resolves lost accuracy when images are distorted instead of blurred [49]. While these fine-tuning methods are promising, Dodge and Karam [11] find that tuning to one type of image quality reduction effect would cause poor generalization to other types of effects. The comparison of image classification between deep neural networks and humans shows similar performance on good-quality images [12]. However, deep neural networks struggle significantly more than humans on low-quality, distorted, and noisy images. One study shows that adversarial perturbations are more prevalent in the Y channel in the YCbCr color space of images than the other two channels, while perturbations in RGB channels are equally distributed [29]. Wu et al. [42] studies the effect of Instagram filters on learning tasks, and introduces a lightweight de-stylization module that predicts parameters used for scaling and shifting feature maps to "undo" the changes incurred by filters.

Researchers have also explored how to improve the robustness of learning algorithms under various image quality degradations. One recent work provides a novel Bayesian formulation for data augmentation [39]. Cubuk et al. [9] proposes an approach to automatically search for improved data augmentation policies. Ghosh et al. [17] analyzes the performance of convolutional neural networks on quality degradations due to compression loss, noise, blur, and contrast, and introduces a method to improve the learning outcome. Another work [21] shows that self-supervision techniques can be used to improve model robustness and exceeds the performance of fully supervised methods. A recent method, AugMix [22] improves model robustness using data augmentation, where transformation compositions are used to create a new dataset that is visually and semantically similar to the original dataset. AugMix is compared to several other augmentation methods, which comprise Cutout [10], MixUp [45], and CutMix [44]. Gao et al. [14] proposes a technique to re-purpose software testing methods to augment the training data of DNNs, with the objective to improve model robustness. A recent work improves model generalizability by first augmenting the training dataset with random perturbations, and then minimizing worst-case loss over the augmented data [18].

Our work differs from these studies in several regards. First, we simulate adversarial conditions of image factors instead of using commonplace image conditions. Second, we conduct a systematic sensitivity analysis for preparing datasets that are representative of image degradations from multiple factors at various levels. Third, our algorithm can work with the discretized parameter space while generalizing well to the continuous parameter space. Another advantage of our approach is that we can augment the training dataset without the derivatives of the factor parameters, which may not exist or are difficult to compute.

## 3 Background and Setup

**Task**. Our target task is end-to-end steering: given a single image as input (e.g. captured by a front-facing camera on a self-driving car), output a steering angle that drives the car safely on the road [4]. A steering angle of 0 represents the forward direction. We use mean accuracy (MA) to evaluate the task since it represents overall performance under a variety of measures (See Sec. 5).

**Datasets**. We choose four real-world driving corpuses as our datasets: Audi [16], Honda [31], Waymo [37], and SullyChen [7]. The Audi dataset is the most recent (2020); the Honda dataset has 100+ long-time driving videos; Waymo includes many environmental conditions such as weather and lightening and is a large dataset (390k frames for perception); and the SullyChen dataset focuses on the steering task and has the longest continuous driving image sequence without road branching. The details of these datasets are provided in Appendix A.3. Other datasets, such as CIFAR-100, for various computer vision tasks are also used to demonstrate the generalizability of our technique.

**Basis perturbation**. We study nine basis perturbations: blur, noise, distortion, three-color (RGB) channels, and hues, saturation, and intensity values (HSV). Blur, noise, and distortion are among the most commonly used perturbations that can directly affect image quality. R, G, B, H, S, V channels are chosen because they are frequently used to represent image color spaces: RGB represent three

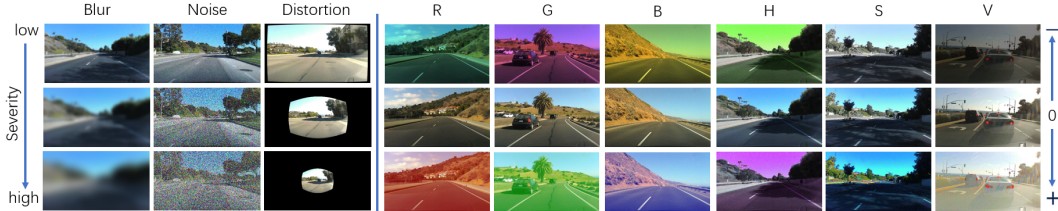

Figure 2: Example images of quality degradation. Left: Five levels of quality reduction using the blur, noise, and distortion effects (three levels are shown). Right: The original images are shown in the middle row. The top row shows examples in the lighter direction per channel for R, G, B, H, S, V, while the bottom row shows examples in the darker direction per channel.

basic color values of an image, while HSV represent three common metrics of an image. Other color spaces such as HSL or YUV have similar properties, hence are excluded.

We use Gaussian blur [2] (parameterized by standard deviation), additive white Gaussian noise (AWGN) (parameterized by standard deviation), and radial distortion [47, 48] (with radial distortion parameters $k1, k2$). For representing channel-level perturbations, we use a linear model: denote the value range of one channel $C$ as $[a_C, b_C]$, in the darker direction, we set $v'_C = \alpha a_C + (1 - \alpha)v_C$, while in the lighter direction, we set $v'_C = \alpha b_C + (1 - \alpha)v_C$, where $\alpha$ is the severity parameter, $v_C$ is the pixel value on clean image and $v'_C$ is the perturbed pixel value. The default values are $a_C = 0$ and $b_C = 255$, with two exceptions. We set $a_V = 10$ to exclude a complete dark image on V channel, and $b_H = 179$ according to H channel definition. See examples in Fig. 2 and detailed description in Appendix A.2.

**Test scenarios.** While other studies usually test in only one or two scenarios [20], we test the performance of all methods in four scenarios with increasing complexity. Scenario 1: *Base dataset*. Test on the base dataset. Scenario 2: *Single perturbation*. Test on the datasets with perturbations, but each dataset only experiences one level of one perturbation (within blur, noise, distortion, R, G, ,B, H, S, V). Scenario 3: *Combined perturbations*. Test on the datasets with combinations of various perturbations at different levels, and each dataset has different levels of all perturbations. Scenario 4: *Unseen perturbation*. Test on the datasets with unseen perturbations at different levels. Specifically, we use "motion blur", "zoom blur", "pixelate", "jpeg compression", "snow", "frost", "fog" from ImageNet-C [20]. We apply the same perturbations/levels to all images within one dataset. (See details in Appendix A.1).

## 4   Improving Robustness of Learning-based Steering

In this section, we introduce our gradient-free adversarial training method to improve the robustness of learning-based steering using Sensitivity Analysis. Our method is among the first to *treat complex unforeseen perturbations as a functional combination of multiple simple "basis perturbations"*. As a result, the robustness of a model to several individual perturbations can lead to robustness against combined or unseen perturbations. In addition, *via discretized sensitivity analysis*, our approach is the first to *use FID to enable cross-factor performance comparison*, i.e., making task sensitivity w.r.t different perturbations comparable. Meanwhile, sensitivity analysis helps minimize the discretization level number, thus speed up the adversarial training process.

### 4.1   Basis Perturbations

There are various types of image corruptions due to different environmental effects or sensor variations, and often these corruptions can be approximated using a combination of basis perturbations, e.g., snow/frost may be simulated using water drop corruptions, Gaussian blur, and image whitening. Inspired by the concept of *basis function*, we explore the possibility of training a model on a few "basis perturbations" and the resulting model is robust against complex perturbations.

Our basis perturbation contains "basis" representations of the color space in RGB and of image metrics in HSV, which span color and intensity/brightness spaces, respectively. Blur and noise are designed to produce low and high frequency corruptions, while distortion captures the 2D positional transformation that may appear from small vibrations and motions of a camera. See comparison of different basis perturbations in Appendix. A.2. We also introduce two metrics to evaluate how well a training method can enable a model to perform on combined perturbations and previously unseen perturbations when learning only on single perturbations. Using basis perturbations to represent

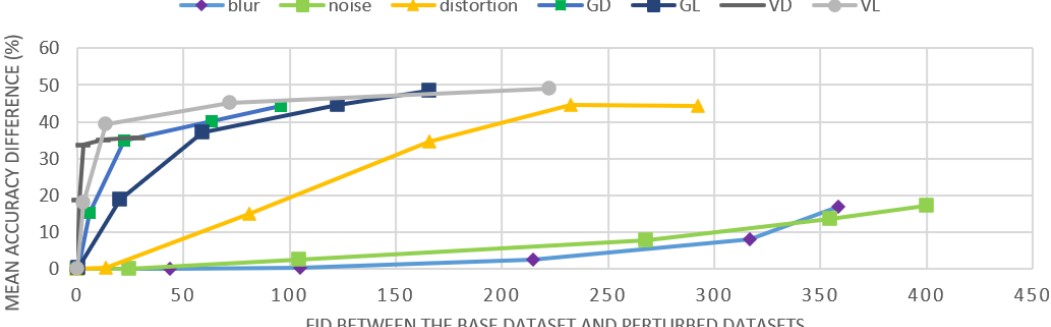

Figure 3: The relationship between FID and MA differences. GD/GL denotes G channel in darker/lighter direction, and VD/VL denotes V channel in darker/lighter direction, respectively. Sensitivity is represented by the first-order derivative of the curve. Note that the values in the near-zero FID range (i.e., $< 50$) are more commonly found in real-world scenarios.

image corruption, we then design an algorithm to train neural networks to cope with quality-degraded images as input for autonomous driving.

## 4.2 Sensitivity Analysis

We use an adversarial training process in which we optimize corruption parameters to maximally disrupt model performance. If this was done using gradient optimization, it would require us to differentiate through the corruption operator to update the parameters describing the corruption. Not all gradients of basis perturbations can be easily derived, and corruptions such as distortions may have parameters with non-differentiable effects. Rather than rely on gradient optimization, we introduce an simple, discretized gradient-free method. We use Sensitivity Analysis (SA) to evenly discretize the space of parameters; we choose a discrete subset of values for each parameter that are equally "far apart" in terms of their impacts on the classifier. This is achieved using the Fréchet Inception Distance (FID) [23] to perform SA across a range of corruption types.

SA is commonly used to understand how the uncertainty of the model output (numeric or otherwise) can be apportioned to different sources of uncertainty of the model input [32, 33]. Here, we use SA to study how distortions to model input (blur, noise, distortion, and RGB/HSV brightness shifts) can be apportioned to model output. We also use SA to quantify the level of degradation caused by an corruption, and prepare datasets that are representative of image qualities at various degradation levels.

We adopt the Fréchet Inception Distance (FID) [23] as a unified metric for our SA (The reasons for adopting FID and a quantitative comparison between FID and other metrics such as L2 norm can be found in Appendix A.4). In addition, we focus on the changes in model performance according to the changes in input and define the sensitivity as the first-order derivative of MA w.r.t FID:

$$sensitivity = \frac{\partial MA^*(R \oplus F(\mathbf{p}))}{\partial FID(R, R \oplus F(\mathbf{p}))},$$

where $MA^*(D)$ is the MA test result on dataset $D$ with the model trained on the base dataset $R$; $FID(A, B)$ is the FID between dataset $A$ and dataset $B$ (where FID is calculated in its standard formulation using a pretrained InceptionV3 network [23]); $F(\mathbf{p})$ is the perturbation with parameter $\mathbf{p}$ (e.g., the standard deviation of the Gaussian kernel), and $D \oplus F$ denotes the dataset obtained by applying perturbation $F$ to $D$.

Starting from empirically-selected parameters of each factor, we generate perturbed datasets and compute their corresponding MA using the trained model on $R$ (see numeric results in Appendix A.9). We then map the MAs and their corresponding parameter values into the FID space. By leveraging this new MA-FID space, we aim to minimize the number of sampled parameters for each perturbation for efficient training (see Sec 4.3), while preserving similar parameter curves. At a high level, we sample points more densely in the value range that has high sensitivity (closer FID values between sample points), while sampling points sparsely in the value range that has low sensitivity (See specific equation in Appendix. A.4). Examples of the resulting images are shown in Fig. 2 (more in Appendix A.1). Detailed descriptions of the final perturbed datasets are provided in Appendix A.2.

The final MA differences in the FID space are visualized in Fig. 3 (for blur, noise, distortion, and G and V channels; see the entire figure in Appendix A.4). We first observe that learning-based steering is more sensitive to the channel-level perturbations (i.e., R, G, B, H, S, V) than the image-level perturbations (i.e., blur, noise, distortion). Second, the task is least sensitive to blur and noise but most sensitive to the V channel (the intensity value). Third, for the same color channel, darker and lighter levels appear to have different MAs at the same FID value. Compared to other "learning to steer" studies [38, 46], our method is the first to transfer perturbations from multiple parameter spaces into one space to enable cross-factor comparison.

## 4.3  Gradient-Free Adversarial Training

Our training process consists of two stages. In the first stage, we use iterative min-max training. At each iteration, we first choose one dataset from the datasets (with different quality degradations) of each basis perturbation (i.e., R, G, B, H, S, V, blur, noise, and distortion) to minimize validation MA, then we merge the nine chosen datasets with the base dataset to train our model while maximizing MA. The first stage stops when a pre-specified number of iterations is reached or the MA loss is below a certain threshold. Our method resembles conventional adversarial training: we improve model robustness by training the model to maximize accuracy using the base dataset plus the perturbed datasets with the minimum accuracy. The loss function is the following:

$$\max_{\theta} \min_{\mathbf{p}} MA(\theta, U_{\mathbf{p}}),$$

where $\mathbf{p}$ represents the union of all parameter levels of all basis perturbations; $\theta$ denotes the model parameters; and $U_{\mathbf{p}}$ is the training dataset. Furthermore, we add a branch in the frequency space to the backbone network to address frequency-related perturbations such as blur and noise (See details in Appendix. A.5). The effectiveness is demonstrated in our ablation study (see Table 1).

The second stage is to boost "clean" accuracy, where the model is fine-tuned solely on the base dataset. To maintain the performance on perturbed datasets, we terminate this stage if the overall validation accuracy (on both clean and perturbed datasets) start to decrease; otherwise, we continue this stage until it reaches the maximum number of iterations or the MA loss decreases to our expected threshold. See Algorithm 1. (Detailed explanations and the ablation study of the second stage are provided in Appendix A.6.)

---

**Algorithm 1:**  Improve robustness of learning-based steering

---

**Result:** a trained model parameterized by $\theta$
**Pre-processing:**
Conduct sensitivity analysis and discretize the parameters of $n$ factors into their corresponding
  $l_{i=1,...,n}$ levels (Sec. 4.2)
Generate new datasets for each factor with the discretized values from the base dataset $R$ (Sec. 3
  Basis Perturbation): $\{D_{i,j}\}_{i=1,2,..,n,j=1,2,..l_i}$
**Initialization:**
Initialize $t = 0$, the maximum number of iterations $T$, the number of epochs $k_1$ and $k_2$, and
  model parameters $\theta^{(0)}$
**First stage:**
**while** $t \leq T$ **do**
  | For each factor, select $D_{i,p_i}$ that can minimize the validation MA, where
  |   $p_i = \arg\min_j MA(\theta^{(t)}, D_{i,j})$
  | Merge all selected datasets $U_{\mathbf{p}} = (\bigcup_{i=1}^n D_{i,p_i}) \bigcup R$
  | Train the network for $k_1$ epochs and update $\theta^{(t+1)} =$ train($\theta^{(t)}, U_{\mathbf{p}}, k_1$) to maximize
  |   $MA(\theta^{(t+1)}, U_{\mathbf{p}})$
  | Break if stop conditions are met; otherwise set $t = t + 1$
**end**
**Second stage:**
Train the network with $\theta^{(final)} =$ train($\theta^{(T)}, R, k_2$) and validate the network on $U_{\mathbf{p}}$ (for $k_2$
  epochs or early break if conditions are met)

---

Our method offers several advantages: 1) the training data is augmented without re-train the model, thus improving efficiency; 2) it provides the flexibility to generate datasets at various discretized levels

of the factor parameters; 3) it does not require the derivatives of factor parameters; other methods that optimize factor parameters in the continuous space require computing derivatives, which can be difficult (e.g., for distortion); 4) it generalizes to not only unseen parameters of individual factors but also the composition of unseen parameters of multiple factors; and 5) it is easy to implement and can be readily integrated with other frameworks such as object detection, classification, and regression.

## 5 Experiments and Results

**Setups.** All experiments are conducted using one Intel(R) Xeon(TM) W-2123 CPU, two Nvidia GTX 1080 GPUs, and 32G RAM. We use the Adam optimizer [24] with learning rate 0.0001 and batch size 128 for training. The maximum number of epochs is 2,000. The dataset setup is explained in Sec. 3. We use the maximum MA improvement (MMAI), the average MA improvement (AMAI), and mean Corruption Errors (mCE) [20] as the evaluation metrics.

**Backbone.** We choose the model described in [4] as the main backbone. We select this model as it has been used to steer an autonomous vehicle successfully in both real world [4] and virtual world [25]. In addition, six other networks are tested to show the generalizability of our method.

**Evaluation metrics.** We define the accuracy w.r.t a threshold $\tau$ as $acc_\tau = count(|v_{predicted} - v_{actual}| < \tau)/n$, where $n$ denotes the number of test cases; $v_{predicted}$ and $v_{actual}$ indicate the predicted and ground-truth value, respectively. We compute mean accuracy (MA) as $\sum_\tau acc_{\tau \in \mathcal{T}}/|\mathcal{T}|$, where $\mathcal{T} = \{1.5, 3.0, 7.5, 15, 30, 75\}$ contains empirically selected thresholds of steering angles. Lastly, we use the maximum MA improvement (denoted as MMAI), the average MA improvement (denoted as AMAI), and mean Corruption Errors (mCE) [20] as the evaluation metrics.

| | Scenarios | | | | | | | | | |
|---|---|---|---|---|---|---|---|---|---|---|
| | Clean | Single Perturbation | | | Combined Perturbation | | | Unseen Perturbation | | |
| Method | AMAI↑ | MMAI↑ | AMAI↑ | mCE↓ | MMAI↑ | AMAI↑ | mCE↓ | MMAI↑ | AMAI↑ | mCE↓ |
| Data Augmentation | -0.44 | 46.88 | 19.97 | 51.34 | **36.1** | 11.97 | 75.84 | 27.5 | 7.92 | 81.51 |
| Adversarial Training | -0.65 | 30.06 | 10.61 | 74.42 | 17.89 | 6.99 | 86.82 | 16.9 | 8.17 | 89.91 |
| MaxUp | -7.79 | 38.30 | 12.83 | 66.56 | 26.94 | 16.01 | 72.60 | 23.43 | 5.54 | 81.75 |
| AugMix | -5.23 | 40.27 | 15.01 | 67.49 | 26.81 | 15.45 | 68.38 | 28.70 | 8.85 | 87.79 |
| Ours w/o FS | 0.93 | 48.57 | 20.74 | 49.47 | 33.24 | 17.74 | 63.81 | 29.32 | 9.06 | 76.20 |
| Ours | **2.12** | **49.97** | **23.92** | **37.30** | 33.15 | **22.12** | **54.38** | **33.16** | **13.81** | **61.61** |

Table 1: Performance of different methods against the baseline [4] on SullyChen dataset. We compare the basic data augmentation method (simply combine all perturbed datasets into training), an adversarial training method [36], MaxUp [18], and AugMix [22]. Overall, our method outperforms all other methods (i.e., highest MA improvements and lowest error in mCEs) in practically all scenarios. Notice ours w/o FS (without frequency space) also outperforms other techniques.

**Comparison with different methods.** We compare our method with four other methods: an adversarial training method [36], a basic data augmentation method, MaxUp [18], and AugMix [22], to see the performance improvement over the baseline method [4]. For the basic data augmentation method, we simply merge all perturbed datasets for model training.

From Table 1, we observe that our method outperforms other methods under all metrics in all scenarios: not only on the clean dataset but also on perturbed datasets. Notably, our algorithm improves the performance of "learning to steer" up to 50% in MMAI, while reducing mCE by 60% over the baseline (Scenario 2). Our method also improves the task performance using the combined datasets (Scenario 3) up to 33%. Lastly, when tested on unseen factors (Scenario 4), our algorithm maintains the best performance by 34% in MMAI, while reducing mCE to 62%.

Compared to AugMix [22], our adversarial approach can select the most challenging datasets for training, thus improving model robustness. MaxUp [18] selects only the worst case among all augmentation data, which may lead to the loss of data diversity. In contrast, our method selects the worst cases in *all* perturbation types (i.e., one dataset per factor), thus improving generalizability. Compared to [36], which performs the adversarial process in feature space by perturbing feature statistics, our method is able to utilize vast prior information generated by sensitivity analysis to reduce the search space. Lastly, compared to the basic data augmentation method, which uses all generated data in training, our method selects the most useful data for training, and thus improves computational efficiency while minimizing data generation.

| | Scenarios | | | | | | | | | |
| --- | --- | --- | --- | --- | --- | --- | --- | --- | --- | --- |
| | Clean | Single Perturbation | | | Combined Perturbation | | | Unseen Perturbation | | |
| Method | AMAI↑ | MMAI↑ | AMAI↑ | mCE↓ | MMAI↑ | AMAI↑ | mCE↓ | MMAI↑ | AMAI↑ | mCE↓ |
| AugMix+Nvidia | -0.12 | 40.64 | 10.94 | 76.48 | 25.97 | 16.79 | 64.41 | **22.23** | 5.99 | 84.95 |
| Ours+Nvidia | **2.48** | **43.51** | **13.51** | **67.78** | **28.13** | **17.98** | **61.12** | 16.93 | **6.70** | **80.92** |
| AugMix+Comma.ai | -5.25 | 55.59 | 9.56 | 86.31 | 31.32 | **0.77** | **106.1** | 37.91 | 7.97 | 89.99 |
| Ours+Comma.ai | **0.36** | **62.07** | **15.68** | **70.84** | **38.01** | 0.74 | 108.32 | **42.54** | **12.15** | **77.08** |
| AugMix+ResNet152 | -4.23 | 20.84 | 1.45 | 96.24 | 12.21 | 6.71 | 80.19 | 15.40 | 2.87 | 97.62 |
| Ours+ResNet152 | **-0.96** | **24.29** | **5.19** | **79.76** | **16.05** | **8.02** | **75.16** | **16.58** | **5.33** | **85.68** |

Table 2: Performance improvement of different backbones against the baseline performance using the Honda dataset. Our method outperforms AugMix in most cases. Notice that the methods with ResNet152 do not improve as much as the first two networks (since the ResNet152 baseline already has high performance).

| | Scenarios | | | | | | | | | |
| --- | --- | --- | --- | --- | --- | --- | --- | --- | --- | --- |
| | Clean | Single Perturbation | | | Combined Perturbation | | | Unseen Perturbation | | |
| Method | AMAI↑ | MMAI↑ | AMAI↑ | mCE↓ | MMAI↑ | AMAI↑ | mCE↓ | MMAI↑ | AMAI↑ | mCE↓ |
| AugMix on SullyChen | -5.23 | 40.27 | 15.01 | 67.49 | 26.81 | 15.45 | 68.38 | 28.70 | 8.85 | 87.79 |
| Ours on SullyChen | **1.46** | **49.76** | **22.87** | **40.84** | **33.15** | **22.12** | **54.38** | **33.87** | **13.51** | **62.50** |
| AugMix on Audi | -8.24 | 81.89 | 32.22 | 55.27 | 75.49 | 50.23 | 41.98 | 73.06 | 27.39 | 77.51 |
| Ours on Audi | **5.98** | **97.57** | **48.50** | **10.27** | **87.56** | **62.38** | **25.80** | **77.22** | **32.71** | **39.14** |
| AugMix on Honda10k | -0.12 | 40.64 | 10.94 | 76.48 | 25.97 | 16.79 | 64.41 | **22.23** | 5.99 | 84.95 |
| Ours on Honda10k | **2.48** | **43.51** | **13.51** | **67.78** | **28.13** | **17.98** | **61.12** | 16.93 | **6.70** | **80.92** |
| AugMix on Honda100k | -11.41 | 63.85 | 14.08 | 70.64 | **68.95** | 47.69 | 40.12 | **61.68** | 16.32 | 88.36 |
| Ours on Honda100k | **-2.55** | **67.35** | **19.88** | **53.26** | 65.10 | **48.60** | **36.94** | 51.90 | **18.29** | **72.84** |
| AugMix on Waymo | 18.27 | 45.40 | 23.30 | 59.31 | **22.95** | 16.92 | 66.36 | **57.65** | 29.10 | 55.63 |
| Ours on Waymo | **20.34** | **46.85** | **26.76** | **52.84** | 21.34 | **18.24** | **64.58** | 56.98 | **31.12** | **53.18** |

Table 3: Performance improvement of different datasets with different sizes against the baseline using the Nvidia backbone. Our method outperforms AugMix considerably in most cases.

**Comparison with different backbones.** We test three backbones: Nvidia network [4], Comma.ai network [34], and ResNet152 [19], on the Honda dataset. The results shown in Table 2 indicate that our method outperforms AugMix in most cases. In general, our method achieves better performance on shallow networks than deep networks. But even on very deep networks such as ResNet152, our method achieves more than 5% improvement in all cases, except Scenario 1.

**Comparison on different datasets.** To demonstrate that our method does not overfit a particular dataset, we experiment four independent datasets: Audi [16], Honda [31], SullyChen [7], and Waymo [37]. We use the Nvidia network as the backbone for these experiments. Table 3 shows that our method achieves consistently better performance across all four datasets. Furthermore, our method obtains up to **97%, 87%, 77%** accuracy improvement on the single, combined, unseen perturbations, while achieving **90%, 74%, 61%** relative error reduction in some cases, respectively.

**Comparison on efficiency.** To analyze the efficiency of our method, we visualize the relationship between *training time vs. robustness* in Fig. 4, using the Nvidia backbone [4] and SullyChen dataset [7]. The x-axis is training time (in seconds) and the y-axis represents the overall robustness, i.e., the average accuracy of the 4 test scenarios (clean, single, combined, and unseen perturbations). Our method outperforms other methods with higher accuracy and efficiency, e.g., after 1/10 the training time (2,500 sec), our performance is already better than others' final performance (with 25,000 sec of training time). Our method is more efficient than others due to discretization of the augmentation space and selection of the most effective augmentation data during the sensitivity analysis. Furthermore, we select the hardest data during training to enable the system to learn more efficiently. Notice that although the frequency space increases the training time per epoch (>2x longer, mainly in the *Fast Fourier Transform* process), it can help to obtain better performance after 7,000 sec even with the same amount of training time, compared with our approach without the frequency-space method.

**Performance on other image processing tasks.** To show the generalizability of our method, we test it on classification using CIFAR-100. For a fair comparison, we use the same setting as of AugMix (i.e., same perturbations, training and testing data, backbone, and code). As shown in Table 4, our method consistently outperforms all backbones, reducing **16.3%** to **34.1%** in mean errors. The data

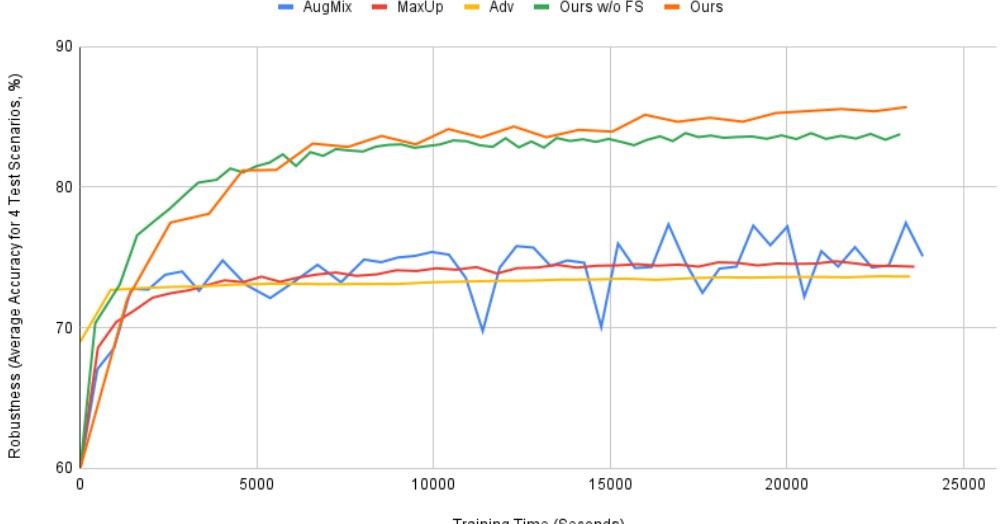

Figure 4: Efficiency comparison between our method and other SOTA methods with the Nvidia Network [4] on the SullyChen dataset [7]. The x-axis is training time (in seconds), while the y-axis displays overall robustness, i.e., the average accuracy of the 4 test scenarios (clean, single perturbation, combined perturbation and unseen perturbation). Our method outperforms other methods with higher accuracy and better efficiency. For example, after 1/10 the training time (2500 sec), our performance is already better than others' final performance (after 25,000 sec of training).

|  | Standard | Cutout | Mixup | CutMix | AutoAugment* | Adv Training | AUGMIX | Ours |
|---|---|---|---|---|---|---|---|---|
| AllConvNet | 56.4 | 56.8 | 53.4 | 56.0 | 55.1 | 56.0 | 42.7 | **25.6** |
| DenseNet | 59.3 | 59.6 | 55.4 | 59.2 | 53.9 | 55.2 | 39.6 | **26.3** |
| WideResNet | 53.3 | 53.5 | 50.4 | 52.9 | 49.6 | 55.1 | 35.9 | **20.5** |
| ResNeXt | 53.4 | 54.6 | 51.4 | 54.1 | 51.3 | 54.4 | 34.9 | **15.4** |
| Mean | 55.6 | 56.1 | 52.6 | 55.5 | 52.5 | 55.2 | 38.3 | **22.0** |

Table 4: Average classification error (in %) across several architectures. Our method is able to reduce mean corruption errors than the previous SOTA methods by 16.3%-34.1% on CIFAR-100-C data.

for other methods come from the AugMix paper [22]. We also test our method on detection in driving. Specifically, we use the Audi dataset [16] and the Yolov4 network [3] as base settings, and implement ours based on Yolov4. The result shows that our algorithm improves robustness in most scenarios (about 3%-5% mAP on average). See Appendix A.7.

**Decoupling Ratio and Generalization Ratio.** We propose two new metrics to evaluate how well a training method can allow a model to perform on combined perturbations and unseen perturbations respectively, while only learning on single perturbations one at a time.

**Definition 5.1** *(Decoupling Ratio) Let $MA(D_1, D_2)$ be the mean accuracy (MA) result of a model trained on dataset $D_1$ and test on dataset $D_2$. Let $P_i^*(D)$ be the ith ($i = 1, ..., n$) perturbation taking dataset $D$ as input and producing a perturbed dataset. Let $NP^*(D) = P_n^*(P_{n-1}^*(...P_1^*(D))...)$ (datasets with combined/nested perturbations), $UP^*(D) = \bigcup_{i=1}^n P_i^*(D)$ (the union of datasets with single perturbations), $D_{tr}$ be the training set, and $D_{te}$ be the test set, then the decoupling ratio is*

$$r_d = \frac{MA(UP^*(D_{tr}), NP^*(D_{te}))}{MA(UP^*(D_{tr}), UP^*(D_{te}))}.$$

$r_d = 1$ means when the model is trained using the union of single-perturbation images, the test performance on combined-perturbation images (not training domain) can be as good as the test performance on the union of single-perturbation images (training domain). Note that different models and training methods may influence the MA function, thus potentially affecting the decoupling ratio.

**Definition 5.2** *(Generalization Ratio) Keeping the definitions of $MA(D_1, D_2)$, $P_i^*(D)$, $D_{tr}$, $D_{te}$, and $UP^*(D)$ as of Decoupling Ratio, let $Q_j^*(D)$ be the jth ($j = 1, ..., m$) perturbation where $P_i^* \neq Q_j^*$, and $UQ^*(D) = \bigcup_{j=1}^m Q_j^*(D)$ (the union of datasets with unseen single perturbations), then the generalization ratio is*

$$r_g = \frac{MA(UP^*(D_{tr}), UQ^*(D_{te}))}{MA(UP^*(D_{tr}), UP^*(D_{te}))}.$$

$r_g = 1$ means when the model is trained using the union of single-perturbation images, the test performance on unseen-perturbation images (not training domain) can be as good as the test performance on the union of known single-perturbation images (training domain). Similarly, different models and training methods may influence the MA function, thus potentially affecting the generalization ratio.

Tested on our dataset ($UP^*(D_{tr})$ is the training set with our basis perturbations, $NP^*(D_{te})$ is the data in Scenario 3, and $UQ^*(D_{te})$ is the data in Scenario 4), our method can achieve a high decoupling ratio $r_d = \frac{0.7463}{0.8836} = 0.84$ and a high generalization ratio $r_d = \frac{0.8291}{0.8836} = 0.94$. We exclude these two ratios for other methods because they did not trained on only single perturbations.

**Effectiveness visualization.** Using the salience map on several combined samples in Fig. 8 shown in Appendix A.8, we demonstrate that our method can help the network to focus on important areas (e.g., the road in front) instead of random areas on perturbed images. We also show t-SNE [26] of feature embeddings from the baseline and our method in Fig. 9 shown in Appendix A.8. As a result, features from our method are more uniformly distributed, indicating the reduction of the domain gaps created by the perturbations, thus improving robustness.

**Benchmarking datasets.** We plan to release our perturbed datasets for benchmarking, which will contain augmented datasets from Audi [16], Honda [31], Waymo [37], and SullyChen dataset [7]. Each will include a base dataset and datasets with five levels of perturbation in blur, noise, and distortion, ten levels of variations in the channels R, G, B, H, S, V, multiple combined perturbations over all nine factors using our implementation, and five levels of each unseen simulated factor, including snow, fog, frost, motion blur, zoom blur, pixelate, and jpeg compression using ImageNet-C. In total, there are 480 datasets and about 26M images. The ground-truth steering angles (or angular velocity of the vehicle) for all images will also be provided for validation, along with the code to generate the perturbed datasets. The parameters for data generation can be found in Appendix A.2.

# 6 Conclusion and Future Work

In this paper, we first analyze the influence of different image-quality attributes on the performance of the task "learning to steer". We study nine image attributes and find that image degradations due to perturbation on these 9 attributes can impact task performance at various degrees. By using FID as a unified metric, we conduct sensitivity analysis in the MA-FID space. Leveraging the sensitivity analysis results, we propose an effective and efficient training method to improve the generalization of learning-based steering under various image perturbations. Our model not only improves the task performance on the base dataset, but also achieves significant performance improvement on datasets with a mixture of perturbations (up to 87%), as well as unseen adversarial examples including snow, fog, and frost (up to 77%).

Our method can be easily extended and applied beyond the set of factors and the learning algorithms analyzed in this study. It can also generalize to analyzing any arbitrarily high number of image/input factors, other learning algorithms, and multimodal sensor data. Lastly, other autonomous systems where the *perception-to-control* functionality plays a key role can possibly benefit from our technique as well. We will release the generated datasets for benchmarking the robustness study of learning algorithms for autonomous driving, as well as the code. Our method currently uses discretization to achieve efficient training, but further optimization for our implementation is possible. Our framework is generalizable to other image factors, learning algorithms, multimodal sensor data, and other perception-to-control tasks.

**Acknowledgement:** This work was supported in part by ARO SI Program, DARPA GARD program, ONR MURI program, Elizabeth S. Iribe Chair, Barry Mersky, Capital One and Pier G. Perotto E-Nnovate Endowed Professorships.

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
