# A   Appendix

## A.1   Dataset Samples

We show different kinds of perturbations in our benchmarks in Fig.5. Specifically, our benchmarks include 9 basic types of perturbations, including Gaussian blur, Gaussian noise, radial distortion, and RGB and HSV channels. Another type of datasets include multiple perturbations, where we create multiple random combinations of the basic perturbations. We also include 7 types of previously unseen perturbations (during training) from ImageNet-C [20], which are snow, fog, frost, motion blur, zoom blur, pixelate, and jpeg compression. For each type of perturbation, we generate 5 or 10 levels of varying intensity based on sensitivity analysis in the FID-MA space.

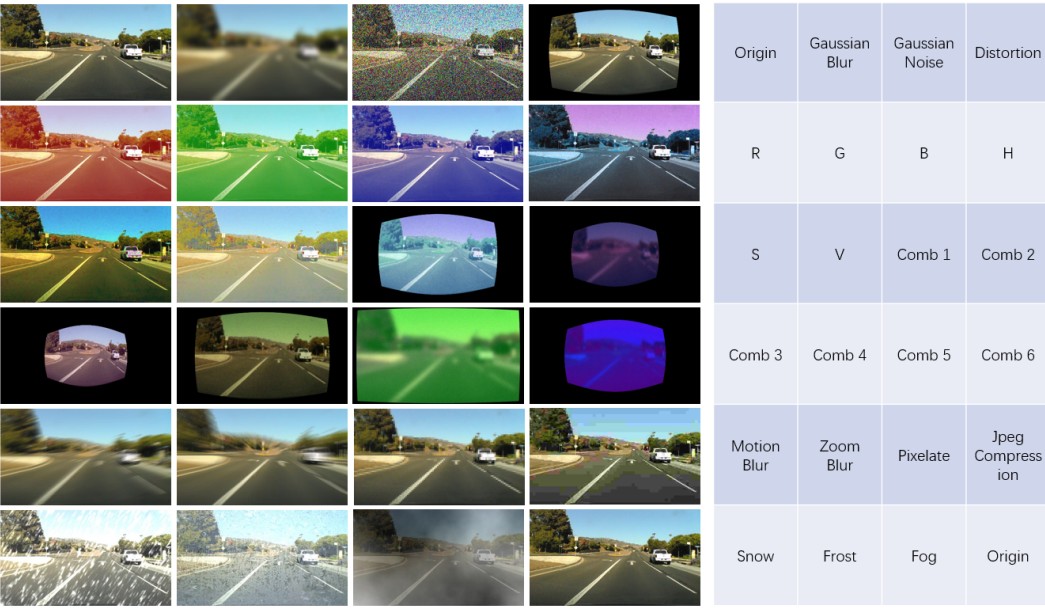

Figure 5: Sample images of our benchmark. We show our benchmark has 22 different types of perturbations. Also, we have 10 levels for R, G, B, H, S, V (5 levels in darker and 5 levels in lighter shades), and 5 levels for each of the other types of perturbations.

We show more image samples of unseen perturbations in Fig. 6.

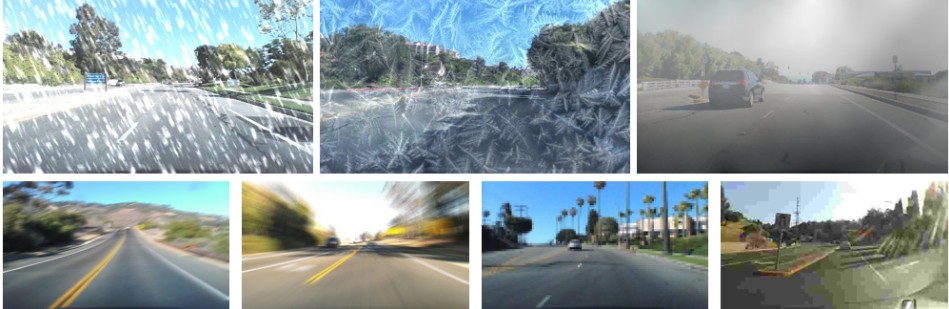

Figure 6: Unseen perturbation examples in our experiments. We use "snow", "frost", "fog" (left to right; first row), and "motion blur", "zoom blur", "pixelate", "jpeg compression" (left to right; second row) from the corruptions in ImageNet-C [20].

## A.2   Perturbed Datasets

We use Gaussian blur [2] (w.r.t standard deviation), additive white Gaussian noise (AWGN) (w.r.t standard deviation), and radial distortion [48] (w.r.t radial distortion parameter $k1, k2$), respectively. For representing channel-level perturbations, we use a linear model: denote the value range of one

channel as $[a, b]$, in the darker direction, we set $v_{new} = \alpha a + (1 - \alpha)v$; in the lighter direction, we set $v_{new} = \alpha b + (1 - \alpha)v$. The default values are $a_C = 0$ and $b_C = 255$, where $C$ represents one channel. To exclude a complete dark image, we set $a_V = 10$ and $b_H = 179$.

In our sensitivity analysis experiments, we first select 10 samples for each of blur, noise, distortion, channel (R, G, B, H, S, V) darker, and channel lighter, then reduce to $n = 5$. We set $n = 5$ since a smaller number like $n = 2$ will decrease the algorithm performance greatly, while a larger number like $n = 8$ will decrease the efficiency dramatically.

The final representative datasets from the sensitivity analysis and used for improving the generalization of the learning task are introduced in the following.

- $R$: the base dataset, Audi [16], Honda [31], or SullyChen [7] dataset;
- $B1, B2, B3, B4, B5$: add Gaussian blur to $R$ with standard deviation $\sigma = 1.4$, $\sigma = 2.9$, $\sigma = 5.9$, $\sigma = 10.4$, $\sigma = 16.4$, which are equivalent to using the kernel (7, 7), (17, 17), (37, 37), (67, 67), (107, 107), respectively;
- $N1, N2, N3, N4, N5$: add Gaussian noise to $R$ with $(\mu = 0, \sigma = 20)$, $(\mu = 0, \sigma = 50)$, $(\mu = 0, \sigma = 100)$, $(\mu = 0, \sigma = 150)$, $(\mu = 0, \sigma = 200)$, respectively;
- $D1, D2, D3, D4, D5$: distort $R$ with the radial distortion $(k_1 = 1, k_2 = 1)$, $(k_1 = 10, k_2 = 10)$, $(k_1 = 50, k_2 = 50)$, $(k_1 = 200, k_2 = 200)$, $(k_1 = 500, k_2 = 500)$, respectively. $k_1$ and $k_2$ are radial distortion parameters, the focal length is 1000, and the principle point is the center of the image.
- $RD1/RL1, RD2/RL2, RD3/RL3, RD4/RL4, RD5/RL5$: modify the red channel of $R$ to darker (D) / lighter (L) values with $\alpha = 0.02$, $\alpha = 0.2$, $\alpha = 0.5$, $\alpha = 0.65$, $\alpha = 1$.
- $GD1/GL1, GD2/GL2, GD3/GL3, GD4/GL4, GD5/GL5$: modify the green channel of $R$ to darker (D) / lighter (L) values with $\alpha = 0.02$, $\alpha = 0.2$, $\alpha = 0.5$, $\alpha = 0.65$, $\alpha = 1$.
- For B, H, S, V channels, we use similar naming conventions for notation as for the red and green channels.
- $Comb1$: $R_\alpha = -0.1180$, $G_\alpha = 0.4343$, $B_\alpha = 0.1445$, $H_\alpha = 0.3040$, $S_\alpha = -0.2600$, $V_\alpha = 0.1816$, $Blur_\sigma = 3$, $Noise_\sigma = 10$, $Distort_k = 17$
- $Comb2$: $R_\alpha = 0.0420$, $G_\alpha = -0.5085$, $B_\alpha = 0.3695$, $H_\alpha = -0.0570$, $S_\alpha = -0.1978$, $V_\alpha = -0.4526$, $Blur_\sigma = 27$, $Noise_\sigma = 7$, $Distort_k = 68$
- $Comb3$: $R_\alpha = 0.1774$, $G_\alpha = -0.1150$, $B_\alpha = 0.1299$, $H_\alpha = -0.0022$, $S_\alpha = -0.2119$, $V_\alpha = -0.0747$, $Blur_\sigma = 1$, $Noise_\sigma = 6$, $Distort_k = 86$
- $Comb4$: $R_\alpha = -0.2599$, $G_\alpha = -0.0166$, $B_\alpha = -0.2702$, $H_\alpha = -0.4273$, $S_\alpha = 0.0238$, $V_\alpha = -0.2321$, $Blur_\sigma = 5$, $Noise_\sigma = 8$, $Distort_k = 8$
- $Comb5$: $R_\alpha = -0.2047$, $G_\alpha = 0.0333$, $B_\alpha = 0.3342$, $H_\alpha = -0.4400$, $S_\alpha = 0.2513$, $V_\alpha = 0.0013$, $Blur_\sigma = 35$, $Noise_\sigma = 6$, $Distort_k = 1$
- $Comb6$: $R_\alpha = -0.6613$, $G_\alpha = -0.0191$, $B_\alpha = 0.3842$, $H_\alpha = 0.3568$, $S_\alpha = 0.5522$, $V_\alpha = 0.0998$, $Blur_\sigma = 21$, $Noise_\sigma = 3$, $Distort_k = 37$

The datasets $Comb1$ through $Comb6$ are generated by randomly sampling parameters of each type of perturbation, e.g. blur, noise, distortion, and RGB and HSV channels, and combining these perturbations together. The parameters listed here are the parameters for the corresponding examples used in the experiment.

We also show the comparison when choosing different basis perturbations in Table. 5. The final set we used are better than other sets (e.g., V channel only, or V channel + B channel + Blur) on clean and unseen perturbation scenarios. Notice we don't compare them on single perturbation and combined perturbation scenarios, as the basis perturbations are different.

## A.3 Dataset details

We use Audi dataset [16], Honda dataset [31], Waymo dataset [37], and SullyChen dataset [7]. Among the autonomous driving datasets, Audi dataset is one of the latest dataset (2020), Honda dataset is one of the datasets that have a large amount of driving videos (over 100+ videos), Waymo dataset includes many environmental conditions such as weather and lightening and is a large dataset

|  | Scenarios | | | |
|---|---|---|---|---|
|  | Clean | Unseen Perturbation | | |
| Basis Perturbations | AMAI↑ | MMAI↑ | AMAI↑ | mCE↓ |
| V channel | -3.33 | 14.61 | 1.64 | 98.54 |
| V channel, B channel, Blur | -0.83 | 15.77 | 2.79 | 95.64 |
| Ours | **2.12** | **33.16** | **13.81** | **61.61** |

Table 5: Performance of different basis perturbations. The set used is better than other sets on clean and unseen perturbation scenarios. We do not compare them on single perturbation and combined perturbation scenarios, as the basis perturbations are different.

(390k frames for perception), and SullyChen dataset is collected specifically for steering task and has a relatively long continuous driving image sequence on a road without branches and has relatively high turning cases.

For Audi dataset [16], we use the "Gaimersheim" package which contains about 15,000 images with about 30 FPS. For efficiency, we adopt a similar approach as in [4] by further downsampling the dataset to 15 FPS to reduce similarities between adjacent frames, keep about 7,500 images and align them with steering labels. For Honda dataset [31], which contains more than 100 videos, we first select 30 videos that are most suitable for learning to steer task, then we extract 11,000 images for Honda10K and 110,000 images for Honda100K from them at 1 FPS, and align them with the steering labels. For SullyChen dataset [7], images are sampled from videos at 30 frames per second (FPS). We then downsample the dataset to 5 FPS. The resulting dataset contains approximately 10,000 images. All of them are then randomly split into training/validation/test data with an approximate ratio 20:1:2. For Waymo dataset [37], we use the data from the perception part directly, which is already split into train/validation/test folders (exclude the domain adaptation data). The training set is about 163k frames, while the test set is about 33k frames. The Waymo dataset doesn't contain steering labels directly, but it contains angular velocity data from IMU. Instead of predicting steering angle, we predict the angular velocity w.r.t. the gravity axis. The only modification is we scale up the angular velocity to meet the range of steering angle, to make it a similar regression problem as steering regression.

There are several good autonomous driving datasets, but not all of them are suitable for the end-to-end learning to steer task. For example, KITTI [15], Cityscapes [8], OxfordRoboCar [27], Raincouver [40], etc., do not contain steering angle labels. Some well-known simulators like CARLA [13] can generate synthetic datasets, but our work focuses on real-world driving using real images. There are also several other datasets that contain steering angle labels (e.g., nuScenes [5], Ford AV [1], Canadian Adverse Driving Conditions [30], etc), but we didn't use them all because the results on the three datasets we chose can already show the effectiveness of our method.

### A.4   FID-MA and L2D-MA

We adopt the Fréchet Inception Distance (FID) [23] as a unified metric for our sensitivity analysis (instead of using the parameter values of each image factor) for three reasons. First, given the autonomous driving system is nonlinear, variance-based measures would be more effective for sensitivity analysis of the network. FID can better capture different levels of image qualities than the parameter values of each factor, because the correspondence between the parameter values and image quality of each factor is not linear. Second, using FID, we can map the parameter spaces of all factors into one space to facilitate the sensitivity analysis. Lastly, FID serves as a comprehensive metric to evaluate the distance between two image datasets: image pixels and features, and correlations among images—these meaningful factors to interpret the performance of a learning-based task—are all taken into consideration.

We first empirically select $m$ parameter values for blur, noise and distortion perturbation severity, and $2m$ parameter values for R, G, B, H, S, V ($m$ in the darker direction and $m$ in the lighter direction), generate perturbed datasets using these parameter values, and compute their corresponding MA using the trained model on $R$ (see numeric results in Appendix A.9). At this point, we can obtain the relationship between MA and parameter values for each factor, but **the parameter values for each factor are not directly comparable to each other**. Notice for each perturbation, one parameter value can generate one perturbed dataset. Thus, we can calculate the FID between this new dataset and clean dataset. In this way, we map the correspondences between the MAs and the

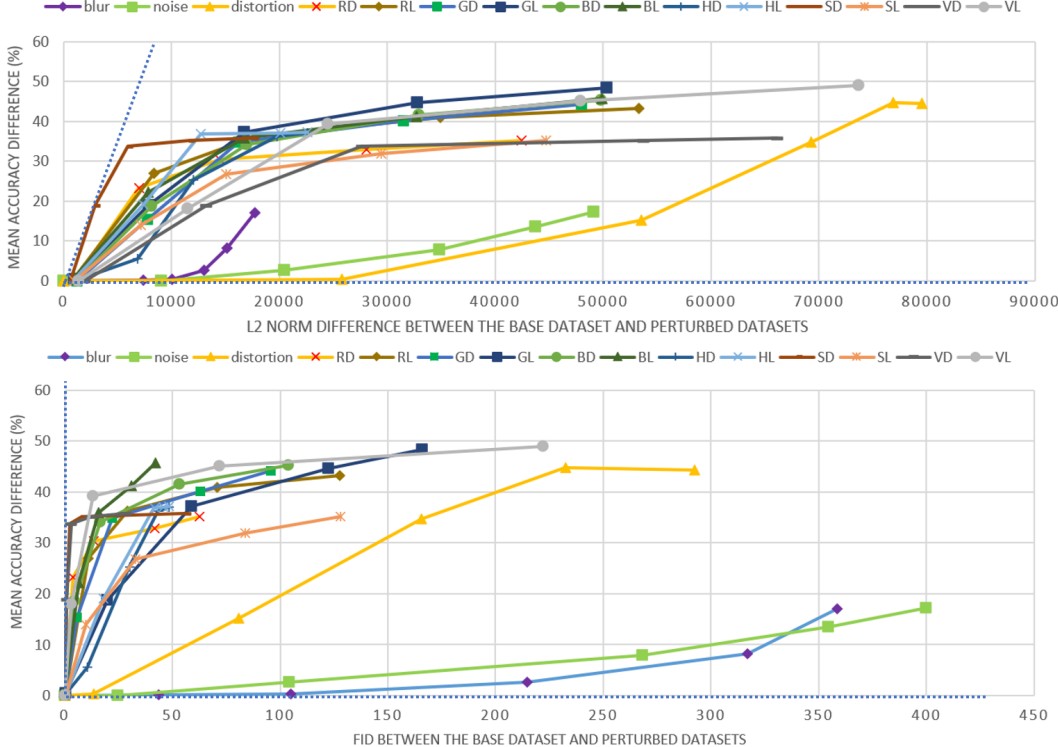

Figure 7: The relationship between L2 norm distance and MA difference (top), and the relationship between FID and MA difference (bottom). The FID space can better capture the difference among various factors affecting image quality better than the L2D space, i.e., the range of the curves' first-order derivative is larger in FID space than in L2D space (see the angle between the two dot lines).

parameter values into the FID space, i.e., FID-MA relationship. By leveraging this new FID-MA space, we can minimize the number of parameter samples for each perturbation, while maintaining a similar FID-MA curve between the sampled dataset and the original one to improve the computational efficiency of training (see Sec 4.3). **A high-level guideline is, in the range that has high sensitivity (See definition in Sec. 4.2) there are denser sample points (closer FID values between sample points), while in the range that has low sensitivity there are sparser sample points**. Specifically, we retain the min and max parameter values, and pick $n - 2$ other parameter values to maximize the minimum 'MA and normalized FID difference' between adjacent sample points (here we assume the FID-MA curve is approximately monotonic):

$$\max_{\substack{Q=\{p_{q_1}, p_{q_2}, ..., p_{q_n}\} \subseteq P \\ q_1 < q_2 < ... < q_n}} \min_i |MA(p_{q_{i+1}}) - MA(p_{q_i})| + \alpha|FID(p_{q_{i+1}}) - FID(p_{q_i})|$$

where $P = \{p_1, p_2, ..., p_m\}$ is the $m$ parameter values we chose in the beginning, $Q = \{p_{q_1}, p_{q_2}, ..., p_{q_n}\} \subseteq P$ is the $n$ parameter values we want to keep, $MA(p_j)$ and $FID(p_j)$ are the mean accuracy value and FID value related to the parameter value $p_j$, and $\alpha$ is the normalization parameter which equals to the reciprocal of the largest FID (within the FID values related to the chosen parameter values). Notice this is not the only criterion to achieve the goal of "sample denser points in high sensitivity range and sample sparser points in low sensitivity range". We also experimented with other criteria, like replacing $|MA(p_{q_{i+1}}) - MA(p_{q_i})| + \alpha|FID(p_{q_{i+1}}) - FID(p_{q_i})|$ with $|MA(p_{q_{i+1}}) - MA(p_{q_i})|$ (MA difference only), which achieves slightly worse results but still works.

We set $m = 10$, try $n = 3, 5, 8$, and find $n = 5$ is the best one ($n = 5$ can lead to similar robustness of the model as $n = 8$ while taking less time, and more robust than $n = 3$). Examples of the resulting

images are shown in Fig. 2 (more in Appendix A.1). Detailed descriptions of the final perturbed datasets are provided in Appendix A.2. We also provide a detailed analysis of MA differences in FID space in Appendix A.4.

The final MA differences in the FID space are visualized in Fig. 7. Since FID aligns different factors into the same space, we can compare the performance loss of all factors at various levels. Notice that the values in the near-zero FID range (i.e., FID< 50) are more commonly found in real-world applications. We first observe that the learning-based steering task is more sensitive to the channel-level perturbations (i.e., R, G, B, H, S, V) than the image-level perturbations (i.e., blur, noise, distortion). Second, the task is least sensitive to blur and noise but most sensitive to the V channel, the intensity value. Third, for the same color channel, darker and lighter levels appear to have different MAs at the same FID values. Compared with other analysis studies on the "learning to steer" task, e.g., [38] and [46], our method is the first to transfer perturbations from multiple parameter spaces into one unified space to enable the cross-factor comparison, e.g., the task is more sensitive to the V-channel perturbation than perturbations in other attributes.

We illustrate the relationship between FID and Mean Accuracy (MA) Difference, and the relationship between L2 norm distance (L2D) and Mean Accuracy (MA) Difference in Fig. 7. As shown in the figure, the FID space can better capture the difference among various factors affecting image quality better than the L2D space, i.e., the range of the curves' first-order derivative is larger in FID space than in L2D space (see the angle between the two dot lines).

## A.5  Frequency Branch of Our Method

When there are frequency-related perturbations (e.g., using Gaussian noise is increasing high frequency component of the image, while using blur operation will reduce high frequency component), a frequency branch can be added to our architecture. Formally, we do a standard 2-D Fourier Transform, and using the absolute value of each complex number and form another channel of the image, thus the input image of the network has 4 channels. The transform equation is:

$$Y_{p,q} = \sum_{j=0}^{m-1} \sum_{k=0}^{n-1} \omega_m^{jp} \omega_n^{kq} X_{j,k}$$

where $\omega_m$ and $\omega_n$ are complex roots of unity:

$$\omega_m = e^{-2\pi i/m}$$
$$\omega_n = e^{-2\pi i/n}$$

$i$ is the imaginary unit. $p$ and $j$ are indices that run from 0 to $m-1$, and $q$ and $k$ are indices that run from 0 to $n-1$.

Notice this frequency branch is optional, our method can already outperform others without this branch. It can help improve the accuracy, but also has limitation: about 1.5x training time. Thus this is an option depends on whether users want the model to be more accurate or more efficient.

## A.6  Second Stage of Our Method

As introduced in Sec. 4.3, the second stage is designed to boost clean accuracy, where the model is fine-tuned solely on clean data. To meanwhile maintain the performance on the perturbed data, we terminate this stage if the overall validation accuracy on clean and perturbed data decreases. Otherwise, this stage continues until it reaches the maximum number of iteration or the MA loss decreases to our expected threshold. In most cases, the network can already learn well the first stage on both clean and perturbed data, thus it will converge very fast in the second stage and do early break without influencing the performance or increasing the training time, as shown in the first row of Table. 6. But we found in some cases, the first stage can make the network learn well on perturbed data, but can not perform on clean data as well as a network trained on only clean data. In this case, adding the second stage can help the network perform better, while doesn't reduce the performance on perturbed data too much. Moreover, our method can perform better than AugMix even without the second stage. See Table. 6 for details.

| | Scenarios | | | | | | | | |
|---|---|---|---|---|---|---|---|---|---|
| | Clean | Single Perturbation | | | Combined Perturbation | | | Unseen Perturbation | | |
| Method | AMAI↑ | MMAI↑ | AMAI↑ | mCE↓ | MMAI↑ | AMAI↑ | mCE↓ | MMAI↑ | AMAI↑ | mCE↓ |
| AugMix on Audi | -8.24 | 81.89 | 32.22 | 55.27 | 75.49 | 50.23 | 41.98 | 73.06 | 27.39 | 77.51 |
| Ours on Audi w/o 2 | 5.86 | 94.95 | 47.78 | 11.83 | **88.42** | 60.31 | 27.18 | 75.16 | **32.91** | **39.04** |
| Ours on Audi | **5.98** | **97.57** | **48.50** | **10.27** | 87.56 | **62.38** | **25.80** | **77.22** | 32.71 | 39.14 |
| AugMix on Honda100k | -11.41 | 63.85 | 14.08 | 70.64 | 68.95 | 47.69 | 40.12 | **61.68** | 16.32 | 88.36 |
| Ours on Honda100k w/o 2 | -7.40 | 66.69 | 17.77 | 58.31 | **69.98** | 47.43 | 37.88 | 58.27 | 17.64 | 80.50 |
| Ours on Honda100k | **-2.55** | **67.35** | **19.88** | **53.26** | 65.10 | **48.60** | **36.94** | 51.90 | **18.29** | **72.84** |

Table 6: Ablation study for the second stage. "w/o 2" stands for "without the second stage training". On Audi dataset, the network is already well trained on both clean and perturbed data, thus the second stage won't make great differences. But on Honda100k dataset, the performance on the clean dataset for the first stage is not well, and adding the second stage can improve the performance on clean data while doesn't influence performance on perturbed data too much. But even without the second stage, our method can perform better than AugMix.

| | Scenarios | | | | | | | | |
|---|---|---|---|---|---|---|---|---|---|
| | Clean | Single Perturbation | | | Combined Perturbation | | | Unseen Perturbation | | |
| Method | AmAPI↑ | MmAPI↑ | AmAPI↑ | mCE↓ | MmAPI↑ | AmAPI↑ | mCE↓ | MmAPI↑ | AmAPI↑ | mCE↓ |
| AugMix | -2.23 | 10.63 | 1.54 | 97.35 | 3.84 | 1.12 | 96.18 | 3.06 | 1.95 | 98.74 |
| Our method | **-1.12** | **16.21** | **3.40** | **95.72** | **7.53** | **4.94** | **94.92** | **5.88** | **2.93** | **96.86** |

Table 7: Performance comparison for detection task against the baseline performance [3]. Our method outperforms the AugMix in all cases, with about 1%-3% mAP improvement on average, while reducing the mCE by 1%-2%.

## A.7 Performance on Detection

We also test our algorithm on the detection task in autonomous driving. We use the Audi dataset [16] (3D Bounding Boxes) and the Yolov4 network [3] as base settings, and then implement our algorithm based on Yolov4. Table 7 shows that our algorithm also improves the model robustness in most scenarios (about 3%-5% mAP improvement on average), and is consistently better than AugMix (about 1%-3% mAP improvement on average).

## A.8 Visualization

We show the saliency map visualization in Fig. 8. Our method can help the network to focus on important areas (e.g., the road in front) instead of random areas on perturbed images.

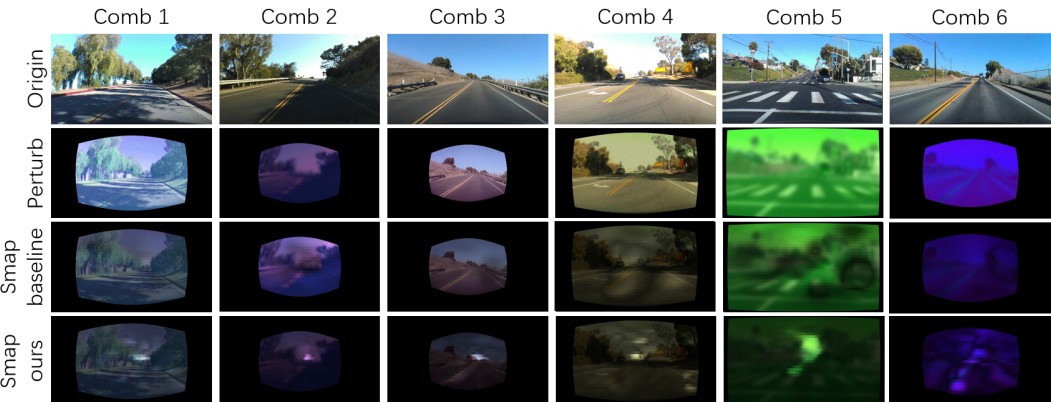

Figure 8: Saliency map samples using the baseline method and our method, where the model is tested on different combinations of perturbations shown as columns. We show the original image, perturbed image with a chosen effect, saliency map of the baseline model, and saliency map of our method from top to bottom rows. Using our method, the network focuses more on the important areas (e.g., road in front) instead of random areas on the perturbed images.

We also show the t-SNE [26] visualization of feature embeddings for the baseline method and our method in Fig. 9. The features from the baseline method are more clustered by color (e.g., the left

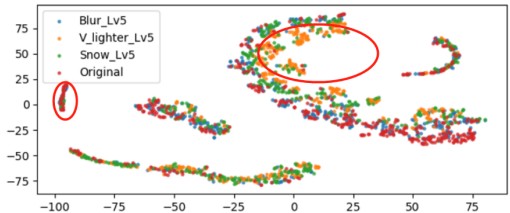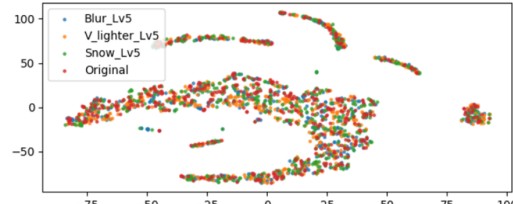

Figure 9: t-SNE [26] visualization for features achieved from networks trained by baseline method (left) and our method (right). The features from the baseline method are more clustered by color (e.g., the left circle in the left image mainly contains red dots, and the right circle in the left image mainly contains yellow dots), indicating there are domain gaps between the perturbed data and original data, while the features from our method are more uniformly distributed, suggesting that our method is able to reduce the domain gaps due to perturbations, i.e., improving the robustness.

circle in the left image mainly contains red dots, and the right circle in the left image mainly contains yellow dots), indicating there are domain gaps between the perturbed data and original data; while the features from our method are more uniformly distributed, suggesting that our method is able to reduce the domain gaps from perturbations, i.e., improve the robustness.

## A.9 Experiment data

To quantify our results, we collected mean accuracy (MA) measurements from each experiment, for each pairwise factor and level across methods. Table 8 shows the mean accuracy measurements for blur, noise, and distortion factors. The same is of table 9, where mean accuracy is measured across levels of RGB or HSV color channels, where each channel serves as a single corruption factor. Table 10 presents the MA measurements for scenarios with a combination of factors, and Table 11 presents the MA measurements for scenes with previously unseen factors.

| Method | Factor | L1 | L2 | L3 | L4 | L5 |
|---|---|---|---|---|---|---|
| baseline | blur | 88.2 | 88.1 | 86.1 | 81.2 | 73.3 |
| | noise | 88.3 | 86.0 | 81.4 | 76.4 | 73.2 |
| | distortion | 88.6 | 75.0 | 57.7 | **48.8** | 49.2 |
| ours | blur | 90.6 | 90.6 | 90.3 | 90.7 | 88.6 |
| | noise | 89.7 | 89.2 | 86.1 | 84.2 | 79.4 |
| | distortion | 90.3 | 89.8 | 87.0 | **84.2** | 83.3 |

Table 8: Mean Accuracy of training (in %) using the baseline model and ours, tested on datasets with different levels of blur, noise, and distortion. Levels range from L1 to L5. We achieve up to 35.4% in performance gain (see bold number pair).

| Method | Factor | DL5 | DL4 | DL3 | DL2 | DL1 | LL1 | LL2 | LL3 | LL4 | LL5 |
|---|---|---|---|---|---|---|---|---|---|---|---|
| baseline | R | 53.2 | 55.4 | 57.9 | 65.1 | 87.8 | 87.7 | 61.4 | 52.1 | 47.4 | 45.1 |
| | G | 44.2 | 48.2 | 53.5 | 73.0 | 88.5 | 87.9 | 69.6 | 51.2 | 43.7 | **40.0** |
| | B | 43.0 | 46.8 | 54.3 | 69.7 | 88.2 | 87.7 | 66.2 | 52.5 | 47.1 | 42.6 |
| | H | 51.3 | 52.1 | 63.1 | 82.8 | 88.1 | 88.2 | 69.3 | 51.5 | 51.3 | 51.2 |
| | S | 58.4 | 63.8 | 72.6 | 83.9 | 88.1 | 88.3 | 74.5 | 61.6 | 56.5 | 53.2 |
| | V | 52.6 | 53.2 | 54.6 | 69.4 | 88.5 | 88.4 | 70.4 | 49.1 | 43.2 | 39.4 |
| ours | R | 88.6 | 90.1 | 90.6 | 90.5 | 90.5 | 90.4 | 90.6 | 90.6 | 90.2 | 89.4 |
| | G | 90.0 | 90.6 | 90.6 | 90.5 | 90.5 | 90.4 | 90.5 | 90.6 | 90.3 | **89.9** |
| | B | 89.1 | 90.0 | 90.5 | 90.4 | 90.3 | 90.4 | 90.4 | 90.6 | 90.0 | 89.3 |
| | H | 89.7 | 90.2 | 90.2 | 90.4 | 90.4 | 90.7 | 90.5 | 90.0 | 89.2 | 89.7 |
| | S | 88.9 | 90.0 | 90.4 | 90.5 | 90.6 | 90.6 | 90.7 | 90.7 | 89.7 | 86.9 |
| | V | 87.8 | 89.5 | 90.7 | 90.8 | 90.7 | 90.5 | 90.6 | 90.0 | 84.4 | 76.4 |

Table 9: Mean accuracy (MA) of training (in %) using the baseline model and ours, tested on datasets with different levels of R, G, B, and H, S, V channel values. DL denotes "darker level", which indicates a level in the darker direction of the channel, while LL indicates "lighter level", which indicates the lighter direction, on levels 1 to 5. We achieve up to **49.9%** in performance gain (see bold number pair).

| Method | Comb1 | Comb2 | Comb3 | Comb4 | Comb5 | Comb6 |
|---|---|---|---|---|---|---|
| baseline | 59.7 | 54.0 | 40.9 | **50.0** | 54.0 | 56.3 |
| ours | 73.4 | 68.6 | 70.9 | **83.3** | 86.2 | 65.3 |

Table 10: Mean accuracy (MA) of training (in %) using the baseline model and ours, tested on datasets with several perturbations combined together, including blur, noise, distortion, RGB, and HSV. We achieve up to **33.3%** in performance gain (see bold number pair).

| Method | Unseen Factors | L1 | L2 | L3 | L4 | L5 |
|---|---|---|---|---|---|---|
| baseline | motion_blur | 76.4 | 69.7 | 62.6 | 61.1 | 60.3 |
| | zoom_blur | 85.6 | 83.7 | 81.8 | 80.0 | 78.2 |
| | pixelate | 88.2 | 88.2 | 88.0 | 88.3 | 88.1 |
| | jpeg_comp | 88.4 | 88.0 | 87.4 | 85.4 | 82.2 |
| | snow | 62.8 | **50.7** | 54.9 | 55.5 | 55.3 |
| | frost | 55.8 | 52.1 | 51.7 | 51.7 | 51.2 |
| | fog | 58.7 | 55.0 | 52.4 | 50.8 | 48.1 |
| ours | motion_blur | 88.0 | 86.9 | 84.7 | 81.4 | 78.5 |
| | zoom_blur | 88.3 | 86.9 | 85.5 | 84.1 | 82.5 |
| | pixelate | 90.3 | 90.3 | 89.9 | 90.5 | 90.6 |
| | jpeg_comp | 90.1 | 90.2 | 90.0 | 89.5 | 90.0 |
| | snow | 87.1 | **83.8** | 82.0 | 77.4 | 76.7 |
| | frost | 86.0 | 83.9 | 81.1 | 81.7 | 80.1 |
| | fog | 77.4 | 71.6 | 65.6 | 61.5 | 56.1 |

Table 11: Mean accuracy (MA) of training (in %) using the baseline model and ours, tested on datasets with previously unseen perturbations at 5 different levels. These types of unseen perturbations do not appear in the training data, and include motion blur, zoom blur, pixelate, jpeg compression loss, snow, frost, and fog, on intensity levels L1 to L5. We achieve up to **33.1%** in performance gain (see bold number pair).