# OpenReview forum: "Gradient-Free Adversarial Training Against Image Corruption for Learning-based Steering"
_NeurIPS.cc/2021/Conference — NeurIPS 2021 Poster_

### Official Review · Reviewer_jMnL · 2021-07-14

**Rating:** 7
**Confidence:** 3

**Summary:**

 In this method, a new augmentation method is proposed to boost the robustness of the supervised “learning-to-steer” and classification tasks for autonomous driving. The proposed approach is based on finding a combination of common perturbations that minimizes the prediction accuracy, and the model is trained in an adversarial manner. The method was evaluated on the driving datasets (Honda, Audi and SullyChen) and CIFAR-100.

**Limitations And Societal Impact:**

Some limitations, such as the need to store the augmented data, are included. However, no practical limitations concerning the speed and efficiency of the proposed method are discussed. Authors claim that their method with composite augmentations requires less training examples as opposed to the methods that use single perturbations. However, this has not been shown experimentally. Ideally, a plot with convergence speed of the proposed vs baseline methods should be included in the paper. Also, an additional preprocessing stage is introduced in the paper, as well as the adversarial training stage, which might be time-consuming, and how much it slows down the training compared to other methods must be shown.


**Main Review:**

The paper is generally difficult to read and needs to be restructured (e.g., L185-193 should be in the experiments section). Some abbreviations and names are used without a clear definition or explanation (e.g. it is not obvious to the non-specialist in autonomous driving what “learning-to-steer” or “steering” is in this context).

“Supervised Contrastive Learning”, Khosla et al. is a very relevant paper that should be included in the related work section.

It is not clear and never discussed in the paper whether the FID is computed using the target model that is currently being trained or a fixed pretrained model. If the latter is true, this approach would always result in a single worst-case composite perturbation.
The overall description of the proposed method in section 4.3 is very confusing and lacks structure. For example, L200-201 state that in stage 2, the model is trained only on the clean data; then later in L204-207, it says that in the same stage 2, the model is trained in an adversarial fashion, and both the clean and perturbed data are used. The description in the Algorithm 1 is too vague: what exactly does it mean: “Conduct sensitivity analysis and discretize the parameters of n factors into their corresponding levels”? Or similarly, “Generate new datasets for each factor with the discretized values from the base dataset”? A reference to a section or equation that clearly describes what is done in this stage is needed.  An exact description of how exactly the new dataset is generated based on the sensitivity analysis is not provided in the paper.

The experimental setup is satisfactory and convincing. An appropriate set of metrics and baseline methods is used to support the main claim of the paper. The empirical results on all driving datasets as well as on CIFAR-100 indicate the efficiency of the proposed approach.

The new metrics proposed in L 279-289, seem to be redundant without a supporting analysis or empirical evidence of their correctness for the evaluation of robustness and generalization.


**Time Spent Reviewing:**

3

---

> ### Author Response · Authors · 2021-08-10
> **Rebuttal (Reviewer jMnL)**
>
> We appreciate your valuable feedback.  Please see our response below.
>
> Paper restructuring:
>
> We show the result of the entire process in the experiment result. We place Fig. 3 and its explanation in Sec. 4.2 to help the reader better understand the sensitivity analysis (e.g., line 174 -- 181). For the ''learning-to-steer'' task, we reference the papers that clearly define this task (e.g., ''End-to-end Learning for Self-driving Cars''). But we will revise the paper to restate the problem statement at both places in the final version to make it more clear.
>
> ''Supervised Contrastive Learning'':
>
> This is a great work! However, it's not natural to apply contrastive loss in regression tasks, such as the steering task for autonomous driving, which aims to regress a steering value according to the image, we didn't include it in our comparison. Theoretically our method and other methods we compared (e.g., AugMix) can be applied to any vision-based task. But we would be happy to discuss this work in the final version.
>
>
> Presentation:
>
> There might be a misunderstanding. FID is computed using a fixed pretrained model, but this process is only used in the sensitivity analysis, before training. It's aiming to augment as little additional data as possible, while retaining the most important information. Then, during the training, the min-max process would use the current trained model to validate the validation datasets, thus it will not result in a single worst-case composite perturbation.
>
> We first describe the method from line 195 -- 204 (stage 1 + stage 2), then we summarize the overall design rationale from line 204 -- 207.
> Overall, our method is adversarial, but for stage 2 it's only trained on clean data and validate on both clean and perturbed data.
>
> Sensitivity analysis and discretization are presented in Sec. 4.2.  "Generate new datasets" is general data augmentation, given the severity of each perturbation.  (Note: a similar process is widely used in other works, e.g., AugMix).
>
>
> New metrics:
>
> We propose the new metrics because, to the best of our knowledge, this is the first paper that presents the design of a robustness algorithm that introduces these important and previously unreported properties on 'basis perturbations'. We believe that the concept introduced here is important and warrant additional investigation in the context of deep learning. We show that a model only trained on single 'basis perturbations' can also perform well on combined or unseen perturbations, this can reduce the complexity in multi-factor robustness dramatically. Other prior works have not exploited these properties, thus these new metrics were not applicable. See more detailed experimental results in the Appendix.
>
>
> Efficiency：
> Please see above in the Main Response.
>
>
> Convergence speed plot:
>
> We'd be happy to provide such a plot in the final version of the supplementary document,
> in addition to the performance numbers on the training time.

---

> > ### Comment · Reviewer_jMnL · 2021-08-25
> > **follow-up comment**
> >
> > The idea proposed in this paper is technically sound, and the results are promising, but the presentation is lacking at this point.
> > I am willing to change my rating to accept, but several modifications to the paper must be made before the end of the discussion period in order for it to conform to the standards of a NeurIPS submission.
> > 1. The method section should be majorly rewritten. In the current state, the description of the algorithm is too vague and needs clarification. Each step that is described with text, e.g. "Perform sensitivity analysis", must be pointed to the paragraph or formula that describes the step *accurately*.
> > 2. The section describing sensitivity analysis must be rewritten clearly, including all the details, such as using a fixed model to compute FID.
> > 3. Although authors refer to another paper that defines the task, it is not expected that all readers of  NeurIPS read the aforementioned paper (if authors expect that then they should have applied their paper to a specialized venue devoted to autonomous driving, not to a general ML conference). Therefore, a clear definition of a task must be included in the main paper.

---

> > > ### Author Response · Authors · 2021-08-27
> > > **Follow-up Rebuttal**
> > >
> > > We very much appreciate your suggestions to help further improve the exposition of this manuscript. We will perform the following edits as you suggested.
> > >
> > > 1. We treat Section 4 as a self-contained Method section:  "sensitivity analysis” (SA) in Section 4.2 and  "adversarial training" in section 4.3. To further improve the clarity, in Section 4.3, we will add a reference to Section 4.2 as needed (line 174~181 is the specific SA step description), and make descriptions for each part clearly (e.g., which step for stage 1, which step for stage 2, what's for the entire process, etc.).
> > >
> > > 2. The entire sensitivity analysis is performed during the preprocessing, so FID can only be computed by a pretrained model. We will make this point more clearly in the revised version. We will also add more details about sensitivity analysis in the appendix to clarify any potential ambiguity.
> > >
> > > 3. We will add an additional brief description of the task to make the definition more clear in the revised version.
> > >
> > > We plan to finish revising the paper as stated above soon. If you have any other suggestions, please let us know and we will improve the exposition further. Thank you again for helping to improve the final exposition of this paper.
> > >
> > > Update: Here's the link for the anonymous revised paper: https://drive.google.com/drive/folders/1jnvi9r3WZfxMNBS3hoKPzc8eBi4Qgmmk?usp=sharing
> > >
> > > Best,
> > >    Authors

---

> > > > ### Comment · Reviewer_jMnL · 2021-09-04
> > > > **rating update**
> > > >
> > > > I would like to thank the authors for their effort to make the Method section more clear and sound. I changed my rating to Accept.

---

> > > > > ### Author Response · Authors · 2021-09-04
> > > > > **THANK YOU**
> > > > >
> > > > > We would like to thank the reviewers for all the excellent suggestions and comments that help further improve the exposition of this paper.
> > > > >
> > > > > best,
> > > > >    Authors

---

### Official Review · Reviewer_cGjr · 2021-07-20

**Rating:** 7
**Confidence:** 3

**Summary:**

This paper introduces an adversarial training framework to help autonomous driving classification systems defend against image corruptions caused by internal and external factors. Motivated by the complex application scenario of auto-driving, the authors study the influence of low-quality images on auto-driving tasks. The authors employ the adversarial training concept of adversarial attack to stimulate image distributions in real-world image corruption to better train auto-driving models.

**Ethical Concerns:**

N.A

**Limitations And Societal Impact:**

The limitations are listed in the main review.

**Main Review:**

This paper proposes an effective gradient-free adversarial training framework for improving the robustness of autonomous driving. The framework will generate a perturbed dataset to stimulate the real-world image corruption distribution and better train auto-driving models. The experimental results show that the proposed method can boost both the robustness and accuracy of the learning to steer models.

Moreover, the authors apply FID to measure the distribution gap between the generated datasets and the real-world images.  Therefore, the authors also propose a comprehensive robustness evaluation standard for learning to steer models.

Good things of this paper:

(a)This paper is well-organized and easy to follow. The pipeline of figure 1 is helpful to understand the proposed research method.

(b) The proposed method has been clearly stated and is easy to be implemented. The authors will also release their codes.

(c) Extensive experiments demonstrate the effectiveness of the proposed methods. The improvement is significant.

Questions of this paper:

(a) The proposed method can be robust to the unseen image corruptions(snow, fog and frost). However, will the proposed training framework also be robust to gradient-based adversarial examples? If no, can the proposed framework add gradient-based adversarial examples for robust training?

(b)In algorithm1, the finding of p is the augment to minimize the MA; how to select p without gradient information?



**Time Spent Reviewing:**

6

---

> ### Author Response · Authors · 2021-08-10
> **Rebuttal (Reviewer cGjr)**
>
> We appreciate your valuable comments, and answer your questions in order.
>
> (a) Robust to gradient-based adversarial examples
> This is a great question. Currently, our method is not designed to handle gradient-based adversarial examples, but rather to handle naturally occurring corruptions. But notice the gradient-based adversarial examples require the perturbation function to be differentiable, while our method can handle the case that the perturbation function is not differentiable, because our method only needs the forward pass to do the sensitivity analysis and training.
> For the second question, yes, this is possible, and it is what we are also working on.
>
> (b) We select p based on the validation accuracy on the perturbed datasets using the latest trained model (See Line 196 in Sec. 4.3).

---

### Official Review · Reviewer_ss88 · 2021-07-22

**Rating:** 5
**Confidence:** 3

**Summary:**

The proposed work uses a concept similar to adversarial training to improve robustness against image corruptions. The authors consider different basis perturbations which can account for different complex perturbations and vary their parameters during the min-max procedure to perform adversarial training.  Empirical results are shown on multiple datasets and different tasks such as classification, detection.

**Ethical Concerns:**

There are no ethical concerns with the paper.

**Limitations And Societal Impact:**

Limitations and societal impact are addressed in the paper.

**Main Review:**

Strengths:

— The proposed method is intuitively well motivated and empirical results are shown on multiple datasets and tasks to show the effectiveness of the approach.

— The idea of breaking down complex perturbations into different basis to simplify the problem is interesting.

— The method is shown to perform better against unseen corruptions as well.

Weaknesses:

— Due the min-max nature of the training, the method seems to be computationally expensive.

— The presentation of the paper can be significantly improved.

1. Precise definitions of MMAI and AMAI would have made the metrics easier to understand.
2. A formal description of the second stage which relies on the frequency component is missing from section A.5 in the appendix.
3. Fig 6 from A.4 can be presented differently. It is difficult to distinguish between different curves.

— Other baselines such as the Data Augmentation and Adversarial training considered in Table 1, are not considered for the detection task.

— It is mentioned that 5 different levels of perturbations are applied for each type of corruption and other levels decreased performance. An experiment which illustrates this effect and explanation for this would improve understanding.

Additional comments:

— Looking at Figure 7 in Appendix A.7, it is difficult to understand where the saliency map of the baseline model is focussed on. It also seems that the proposed approach is focussing on the center potion of the image. Other qualitative examples might be helpful here.

— An experiment where all the generated datasets are used in training, rather than choosing a dataset based on MA, serves as a baseline. It would be interesting to see if the improvement for the proposed approach is due to the additional data or min max procedure.

— An experiment where different basis perturbations are considered during training( more or less in number, different subset) serves as an ablation and would motivate the use of the selected perturbations.

**Time Spent Reviewing:**

7

---

> ### Author Response · Authors · 2021-08-10
> **Rebuttal (Reviewer ss88)**
>
> Thank you for valuable feedback.  We address issues raised and additional comments mentioned below.
>
> WEAKNESS
>
> Efficiency
> please see above.
>
> Presentation
> (1) MMAI and AMAI are defined in Sec.5 on ''Evaluation Metric''.  We first define the mean accuracy (MA), then define the maximum MA improvement (MMAI) and average MA improvement (AMAI). Our metrics use relative improvement over a baseline instead of raw accuracies because accuracy-based metrics tend to be strongly dominated by performance on the few hardest tasks and ignore the small accuracy improvements in easier tasks with higher baseline performance.  Note that our approach is analogous to the mCE metric, which comes from AugMix and is also a relative metric that uses a baseline.
> (2) The frequency branch process is a standard 2D FFT process followed by a network with the same architecture as the image branch, thus we only briefly describe it in Sec.4.3, given the page limit, and show ablation study in Table 1.   Appendix A.5 is about the second stage training, not for the frequency branch.
> (3) Fig.6 contains all the curves, as it's aiming to show the overall trend of the curves, as described in A.4.
> For detailed analysis, please refer to Fig.3, which is easier to see the trend, while Fig. 6 is left in the appendix given the page limit.
>
> Other baselines
> The adversarial training is already contained in Table 4. See the "Adv Training" column.  Note that our original submission already contains comparisons between our method and
> 7 other SOTA methods.  If the reviewer has another 'baseline' in mind that would outperform the seven we used for comparison, we would do our best to include it in the final version.
>
> Performance decrease experiments
> We already included this experiment data in A.8, i.e., how performance will decrease for each level of each type of corruption.
> In fact, this is a required step in our method, as part of the sensitivity analysis that would minimize the amount of
> training and storage requirements.
>
>
> ADDITIONAL COMMENTS
>
> Saliency map
> In Fig.7, the baseline model can not focus on the right part thus it can not get good results. Our method looks like focusing on the center point because of the natural property of autonomous driving data. Also, notice that though the 6 cases have different distortions, all of them can somehow locate the front road, which shows our method is not remembering a fixed position on the image, but focusing on the right content in the image.
>
> Additional data or min-max procedure
> Our submission showed results of (1) original method: no additional data, no min-max procedure; (2) data augmentation: additional data, no min-max procedure; (3) our method: additional data, min-max procedure. See results in Table 1 (Sec. 5). Note: we did not include a combination of "no additional data, min-max procedure", because if there are no (explicit or implicit) data augmentations, there's no meaningful way to apply the min-max approach for robustness.
>
> Different basis perturbations
> Actually, we did many experiments for different sets of 'basis perturbations' during the exploration phase of this project.  We would be happy to provide those additional results in the final version of the supplementary document, though they might be 'too much information' for readers. In general the most valuable lesson learned about the basis perturbation selection is summarized in Sec. 4.1.
> It is possible to choose other sets of 'basis perturbations' that span the same space, as long as these sets of basis perturbations can
> sufficiently capture the color representations (e.g. RGB vs. CMY/CMYK), variations in brightness, contrast, and shading (i.e. hue, saturation, intensity), and uncertainty/errors/quantization in sensors (like cameras) and image processing (e.g. noise, blur, distortion).

---

> > ### Comment · Reviewer_ss88 · 2021-09-01
> > **Response to Rebuttal**
> >
> > Thanks to the authors for addressing some of the concerns. The other baselines that were mentioned under weaknesses for the detection task whose results are shown in Table 6 of the appendix and does not include all the baselines. Table 4 of the main paper consists of experiments with different backbones for classification task which was not the experiment under consideration.  The authors are also encouraged to include the experiments with different basis perturbations as it provides quantitative comparison with different selections. In L209 of the main paper, the authors include a brief description of  the "frequency space" branch after the introduction of the second stage and no formal description is provided in the main paper or under section A.5 in supplementary. This makes it hard to understand the exact experiment and the authors are suggested to provide more details. For these reasons, I will keep my score unchanged.

---

> > > ### Author Response · Authors · 2021-09-04
> > > **Follow-up Rebuttal**
> > >
> > > Thanks for your comments.  The main idea of our method is to decouple complex combined perturbations into ‘basis perturbations' and generalize to unseen perturbations, with the target domain of autonomous driving and focus on END-TO-END, COUPLED PERCEPTION-CONTROL task of learning-based STEERING. The detection results, different basis perturbation, and frequency space are not the key issues. Please see below.
> > >
> > > Here's the link for the anonymous revised paper: https://drive.google.com/drive/folders/1jnvi9r3WZfxMNBS3hoKPzc8eBi4Qgmmk?usp=sharing
> > >
> > > 1. Q: The other baselines that were mentioned under weaknesses for the detection task whose results are shown in Table 6 of the appendix and does not include all the baselines. Table 4 of the main paper consists of experiments with different backbones for classification task which was not the experiment under consideration.
> > >
> > > A: Our target task is the *end-to-end* learning-to-steer task. We show the comparison of other image processing tasks like classification and detection to illustrate the generalization of our method. Given the page limit, we could only provide the full comparison and ablation studies on the primary task of end-to-end steering, which involves both perception and control. We also included CLASSIFICATION results in the main paper and the DETECTION results in the appendix. We also added a comparison with AugMix for the detection task in the revised version (Table 7), where ours is 1%~3% better, and we can include more additional comparisons in the final version if you are interested. In Table 4, the backbones we used come from the AugMix paper. We conducted our experiments based on their exact settings for complete fairness in comparison.
> > >
> > > 2. Q: The authors are also encouraged to include the experiments with different basis perturbations as it provides quantitative comparison with different selections.
> > >
> > > A: As we mentioned in the previous rebuttal, we did many experiments for different sets of 'basis perturbations' during the exploration phase of this project. We also include these experimental results now in the revised version (Table 5 in Appendix) to illustrate that the chosen basis perturbations perform better than the alternative methods. We would also be happy to provide more additional results in the final version of the supplementary document, but the results will likely be overwhelming to readers while providing little more insights, given the volume of data. In addition, the current supplementary document is already as long as the paper. In general the most valuable lesson learned about the basis perturbation selection is summarized in Sec. 4.1. It is possible to choose other sets of 'basis perturbations' that span the same space, as long as these sets of basis perturbations can sufficiently capture the color representations (e.g. RGB vs. CMY/CMYK), variations in brightness, contrast, and shading (i.e. hue, saturation, intensity), and uncertainty/errors/quantization in sensors (like cameras) and image processing (e.g. noise, blur, distortion), as we stated in our earlier rebuttal text.
> > >
> > > 3. Q: In L209 of the main paper, the authors include a brief description of the "frequency space" branch after the introduction of the second stage and no formal description is provided in the main paper or under section A.5 in supplementary. This makes it hard to understand the exact experiment and the authors are suggested to provide more details.
> > >
> > > A: As mentioned in the previous rebuttal, the frequency branch process is a standard 2D FFT process followed by a network with the same architecture as the image branch, we added a formal description in the revised version (Appendix. A.5). Even without the frequency branch, our method already outperforms other SOTA methods (Table 1).

---

### Author Response · Authors · 2021-08-10
**Efficiency Explanation**

Dear Reviewers: We appreciate your valuable comments. We address a key issue here and clarify the confusion below.

Efficiency: The reviewers are correct that adversarial augmentation methods have the potential to increase training time. A major focus of our paper is on ways to minimize the additional costs of adversarial perturbations so that the user can enjoy the robustness benefits of adversarial augmentation with as little additional computation as necessary. Compared to naive (and expensive) data augmentation / adversarial training, we propose to (1) select a strong subset of the augmented datasets that maximally improve the model's learning without the costs of considering all datasets on each step, (2) we decouple combined perturbations and instead train solely on single ``basis perturbations'', thereby significantly reducing the dimension of augmented image space dramatically,  and (3) utilize sensitivity analysis during preprocessing to optimally bin perturbation parameters into a small number of discrete values that can be searched quickly. This analysis uses FID scores to parameterize the impact and severity of perturbations across 9 image attributes. These pre-processing (and pre-training) steps are not free, but they dramatically speed up the process of searching for optimal perturbations.

Compare to naive data augmentation on combined perturbations, our method offers exponential speedup, while comparing to naive data augmentation on single perturbations, ours can provide linear speedup, w.r.t. the numbers and levels of perturbation. Experimentally, our method is about 10x faster compared to the naive data augmentation on single perturbations.

Comparison with SOTA Methods:  In practice, the convergence rates may change depending on a  specific task or dataset. In our experiments on CIFAR-100-C, our method can reduce 20% mCE compared to the AugMix, while using only about 80\% of training time (i.e. our method converges faster while reducing errors in this experiment, see Table A below for details).
In other experiments like Honda100k for steering, our method achieves slightly better performance with the same training time as AugMix; it further reduces 10\% mCE in about 1.5x of AugMix training time.

Our approach is not completely `free', but runs comparably fast, if not faster, to AugMix with noticeably reduced mCE error rates. For other methods (like data augmentation, adversarial training, MaxUp, etc), each of their overall efficiency is no better than AugMix in our experiments. We are happy to include additional convergence plots of ours vs. these methods to illustrate the efficiency of training (and considerable reduction of training time with reduced error rates, as shown in Table 1) by our approach in the final version.

TABLE A: Efficiency and mCE comparison between AurMix and our method on CIFAR-100-C
Training process                 20%    40%    60%    80%    100%
Training time in sec (AugMix)    746   1492   2239   2985    3732
Training time in sec (Ours)      585   1171   1756   2342    2928
mCE (AugMix)                   60.40  58.30  52.36  45.28   40.35
mCE (Ours)                     53.23  42.55  34.86  25.13   20.50


The current implementation of our approach is not optimized.  Although our method is not as time-consuming as other data augmentation or adversarial training techniques, the efficiency of our system can still be further enhanced using techniques like re-weighting strategy [Ren et al, "Learning to reweight examples for robust deep learning", ICML 2018], dynamic switching for online and offline data generation by combining ours and MaxUp [Gong et al, arxiv 2020]), etc.

---

### Decision · Program_Chairs · 2021-09-27

**Decision:**

Accept (Poster)

**Comment:**

This paper proposes a new approach for adversarial training. The idea seems novel and interesting and the result seems to be promising. The paper presentation and organization should be further improved. The meta reviewer recommends acceptance of this paper.